# Commensal *E. coli* limits *Salmonella* gut invasion during inflammation by producing toxin-bound siderophores in a *tonB*-dependent manner

**Yassine Cherrak***, **Miguel Angel Salazar, Koray Yilmaz, Markus Kreuzer, Wolf-Dietrich Hardt** *

Institute of Microbiology, Department of Biology, ETH Zurich, Zurich, Switzerland

* ycherrak@biol.ethz.ch (YC); hardt@micro.biol.ethz.ch (W-DH)

**Data Availability Statement:** All relevant data are within the paper and its Supporting Information

## Abstract

The gastrointestinal tract is densely colonized by a polymicrobial community known as the microbiota which serves as primary line of defence against pathogen invasion. The microbiota can limit gut-luminal pathogen growth at different stages of infection. This can be traced to specific commensal strains exhibiting direct or indirect protective functions. Although these mechanisms hold the potential to develop new approaches to combat enteric pathogens, they remain far from being completely described. In this study, we investigated how a mouse commensal *Escherichia coli* can outcompete *Salmonella enterica* serovar Typhimurium (*S*. Tm). Using a salmonellosis mouse model, we found that the commensal *E. coli* 8178 strain relies on a trojan horse trap strategy to limit *S*. Tm expansion in the inflamed gut. Combining mutants and reporter tools, we demonstrated that inflammation triggers the expression of the *E. coli* 8178 antimicrobial microcin H47 toxin which, when fused to salmochelin siderophores, can specifically alter *S*. Tm growth. This protective function was compromised upon disruption of the *E. coli* 8178 *tonB*-dependent catecholate siderophore uptake system, highlighting a previously unappreciated crosstalk between iron intake and microcin H47 activity. By identifying the genetic determinants mediating *S*. Tm competition, our work not only provides a better mechanistic understanding of the protective function displayed by members of the gut microbiota but also further expands the general contribution of microcins in bacterial antagonistic relationships. Ultimately, such insights can open new avenues for developing microbiota-based approaches to better control intestinal infections.

## Introduction

*Salmonella* enterica serovar Typhimurium (*S*. Tm) is a foodborne pathogen and a leading cause of non-typhoidal salmonellosis (NTS). NTS is responsible for an estimated 93 million cases of gastroenteritis worldwide, some of which escalate into life-threatening systemic infections [1]. Alongside its implication in diarrheal diseases, the emergence of drug-resistant *S*. Tm strains poses as a major global health concern which has amplified the urgency for alternative treatment options [2,3].

files. Source data used for graphs in all figures can be found in the S1 Data file.

**Funding:** This work was funded by grants from the Swiss National Science Foundation (310030_192567: www.snf.ch/en) and the NCCR Microbiomes (SNF grant number 180575: www.nccr-microbiomes.ch) attributed to W.-D.H. Y.C was supported by an EMBO long-term fellowship (ALTF-24-2020: www.embo.org) and a flexibility grant from the SNF/NCCR Microbiomes (51NF40_180575). The funders had no role in study design, data collection and analysis, decision to publish, or preparation of the manuscript.

**Competing interests:** The authors have declared that no competing interests exist.

**Abbreviations:** C.I, competitive index; c.f.u, colony forming unit; HE, hematoxylin and eosin; LB, lysogeny broth; NTS, non-typhoidal salmonellosis; O.D, optical density; PBS, phosphate-buffered saline; qPCR, quantitative PCR; SPF, specific pathogen-free; T3SS, type-three secretion system; TBDR, *tonB*-dependent receptor.

Our understanding of *S.* Tm invasion and pathogenicity has been greatly advanced using antibiotic pretreated or gnotobiotic mouse models that are permissive to oral infections [4,5]. By alleviating gut colonization resistance, these approaches allow efficient gut infections by *S.* Tm [6,7]. Colonization resistance, identified in the 1950s after observing a significant increase in intestinal *S.* Tm loads upon antibiotic pretreatment, refers to the ability of an unperturbed gut microbiota to act as a natural protective barrier restricting the growth of incoming pathogens [8–10]. Gut commensal bacteria appear to limit pathogen growth in various ways. One of them involves resource limitation, where commensal strains affect the growth of their competitors by depriving them of nutrients necessary for their proliferation. This scenario is exemplified by the commensal *Klebsiella michiganensis* and *Escherichia coli* Mt1B1 strains which were shown to attenuate *S.* Tm growth through the consumption of the sugar-alcohol galactitol [11,12]. In a similar fashion, depletion of *Clostridia* was shown to increase luminal oxygenation favouring *S.* Tm bloom, while the human probiotic *E. coli* Nissle 1917 strain (Mutaflor) has demonstrated an efficient ability to outcompete *S.* Tm via oxygen depletion [13–15]. Besides oxygen intake, *E. coli* Nissle can outgrow *S.* Tm through the acquisition of iron, a crucial metabolite whose availability is restricted during inflammation [16]. This effect was attributed to the energizing TonB protein which facilitates the active uptake of siderophores in the iron-limited inflamed gut. Mechanistically, how *E. coli* Nissle outgrows *S.* Tm using similar and shared siderophores is unknown. As an alternative to consuming common nutrients, members of the gut microbiota can directly compete against pathogenic strains through interference mechanisms such as antimicrobial metabolites or toxins. For instance, commensal *Bacteroides* spp. can produce propionate which dampens *S.* Tm growth in a pH-dependent fashion [17]. In another example, *E. coli* Nissle utilizes low-molecular weight proteins called microcins to specifically target *E. coli* and *S.* Tm competitors in the gut [18]. Despite the promising potential of the gut microbiota as a mean of preventing and treating *S.* Tm infections, only a small number of protective commensal strains have been successfully isolated and thoroughly characterized.

Upon intestinal infection, *S.* Tm employs the type-three secretion systems (T3SSs) encoded within the *Salmonella pathogenicity islands* (*spi*) -1 and -2 to respectively invade the intestinal epithelium cells and survive within tissue phagocytes [19]. The resulting immune activation and inflammation not only further disrupts the gut microbiota and its associated protective function but also liberates growth fuelling metabolites benefiting *S.* Tm as well as members of the *Enterobacteriaceae* family [19–28]. This co-blooming phenomenon arises from a strong metabolic overlap between *S.* Tm and closely related *E. coli* isolates, such as *E. coli* 8178, that were shown to exhibit protective functions [7,15,25]. *E. coli* 8178 is a murine gut commensal with a remarkable ability to limit the growth of *S.* Tm in various mouse infection models [25,29,30]. *E. coli* 8178's capacity to attenuate *S.* Tm invasion can synergize effectively with additional competing strains, and complete clearance of *S.* Tm from the gut of infected animals is achieved when combined with a mucosal vaccination raising secretory IgA [30,31]. However, despite the potential of *E. coli* 8178 in limiting *S.* Tm infection, the mechanism underlying this protective function remains undiscovered. In the present study, we aimed at understanding how *E. coli* 8178 outcompetes *S.* Tm in the murine gut. For this purpose, we employed a *Salmonella* mouse infection model and bacterial genetic approaches to decipher, at the molecular level, how *E. coli* 8178 eliminates *S.* Tm in vivo.

## Results

### *E. coli* 8178-mediated *S.* Tm elimination is triggered by inflammation

The 129S6/SvEvTac mice are known to develop chronic Salmonellosis with persistent infection, making them a suitable model for studying the *S.* Tm attenuating capacity of *E. coli* 8178

[32]. However, *E. coli* 8178 was originally isolated from the microbiota of C57BL/6 mice where it can efficiently outcompete *S.* Tm, and it remains unclear whether *E. coli* 8178-mediated elimination can occur in a different mouse line harboring a distinctly complex gut microbiota [25]. To answer this question, we conducted in vivo experiments using specific pathogen-free (SPF) microbiota-colonized 129S6/SvEvTac mice. Prior to infection, animals were pretreated with a single dose of streptomycin to impair gut colonization resistance and facilitate bacterial invasion (Fig 1A) [6]. Antibiotic-pretreated SPF 129S6/SvEvTac mice infected exclusively with the *S.* Tm SL1344 strain exhibited a high pathogen load throughout the 4 days of the infection, averaging $10^9$ to $10^{10}$ *S.* Tm cells per gram of faecal samples (Fig 1B). Besides the slight decrease (approximately 10-fold) in *S.* Tm loads visible at the early stage of infection (24 h p.i), co-infection with a 1:1 ratio of *S.* Tm and *E. coli* 8178 led to equivalently high gut pathogen luminal loads after 48 h of infection (Figs 1B and S1A). This load displayed a more striking decline (1.000- to 10.000-fold) 72 to 96 h post-infection (Figs 1B and S1A) which aligned with previous observation in C57BL/6 mice and prompted us to investigate the underlying mechanisms [25]. In that perspective and although *S.* Tm was similarly outcompeted in *E. coli* 8178-preinoculated animals (S1B and S1C Fig), we opted for a co-infection experimental setup. We next assessed the effect of *E. coli* 8178 on *S.* Tm-elicited intestinal inflammation. For this purpose, we quantified the level of faecal host lipocalin-2 which serves as a general marker of gut inflammation [33]. *S.* Tm-infected 129S6/SvEvTac mice experienced intestinal inflammation starting from 24 h post-infection peaking 48 h after infection (Fig 1C). This inflammation level remained stable throughout the entire course, in accordance with earlier observations [34]. Apart from the early stages of infection (24 h p.i), the presence of *E. coli* 8178 did not impact the onset of *S.* Tm-elicited inflammation (Fig 1C). This observation indicated that *E. coli* 8178-mediated *S.* Tm competition occurred in the later stages of infection, particularly when inflammation is established (Fig 1B and 1C). Inflammation is associated with drastic immune and metabolic changes affecting bacterial composition and competitive behavior in the gut lumen [18,24,35]. Given the severely elevated level of gut inflammation observed when *S.* Tm began to be outnumbered by *E. coli* 8178, we hypothesized that inflammation might serve as a trigger for such antagonistic interaction. To test this hypothesis, we evaluated the competitiveness of *E. coli* 8178 against an isogenic *S.* Tm mutant that was unable to initiate inflammation. For this purpose, we used a *S.* Tm strain deprived of the T3SSs encoded in the *spi*-1 and *spi*-2 regions (*spi*-1/*spi*-2) [21]. Comparably to C57BL/6 animals, antibiotic pretreated 129S6/SvEvTac mice infected with *S.* Tm *spi*-1/*spi*-2 mutant displayed a high bacterial load averaging $10^8$ cells per gram of faeces 96 h post-infection (Fig 1D) [21]. In this context, no severe inflammation was detectable (S1D and S1E Fig) and although stably colonizing, addition of *E. coli* 8178 did not alter the growth of the *S.* Tm *spi*-1/*spi*-2 mutant (Figs 1D, S1E and S1F). To confirm the contribution of inflammation in *E. coli* 8178's protective behavior, we conducted a competitive infection experiment with *E. coli* 8178 and the *S.* Tm *spi*-1/*spi*-2 mutant in mice where intestinal inflammation was restored by the addition of the WT *S.* Tm strain (S1E Fig). In the resulting triple-infected animals (WT *S.* Tm + *spi*-1/*spi*-2 *S.* Tm + *E. coli* 8178), the WT and *spi*-1/*spi*-2 *S.* Tm strains were both outcompeted by *E. coli* 8178 (Fig 1D). We thus concluded that inflammation acts as a triggering factor for *E. coli* 8178 to exert its competitive advantage against *S.* Tm in the gut.

## *E. coli* 8178 relies on *tonB* to outcompete *S.* Tm in the inflamed gut

Numerous colonization resistance mechanisms between closely related strains were reported to rely on nutrient utilization [11,12,16,36]. To identify the metabolic pathway responsible for *S.* Tm growth reduction, we designed a barcoded mutant pool of *E. coli* 8178. For this purpose,

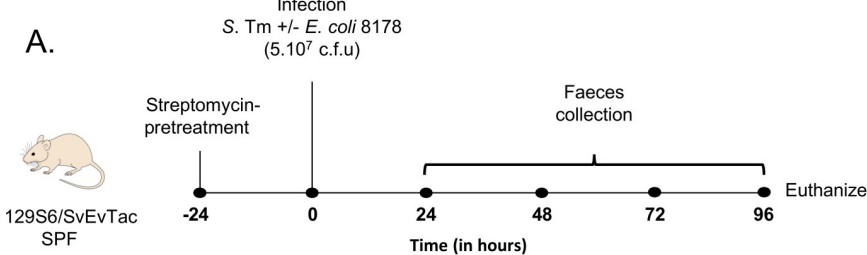

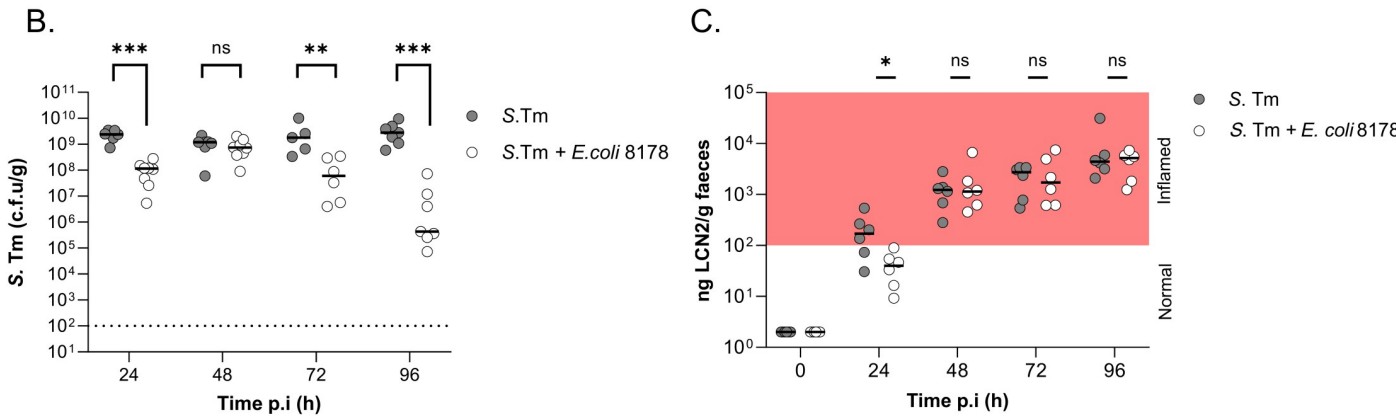

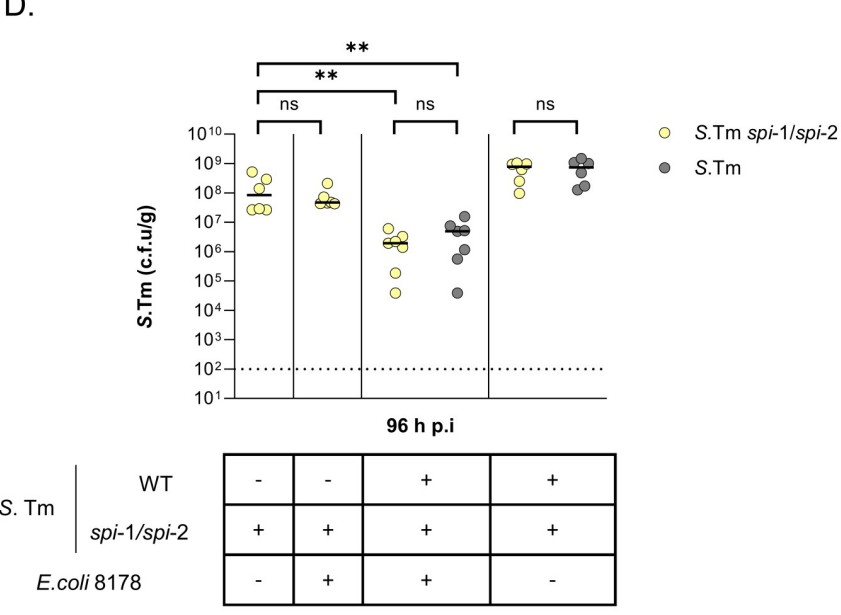

**Fig 1. Inflammation triggers *S.* Tm elimination by the commensal *E. coli* 8178.** (**A**) Experimental scheme. Streptomycin-pretreated SPF 129S6/SvEvTac mice were infected with either *S.* Tm or an equal mixture of *S.* Tm + *E. coli* 8178. The *E. coli* 8178 and *S.* Tm loads were determined by selective platting from faecal samples collected 24, 48, 72, and 96 h after infection. Mice were euthanized at day 4 p.i. (**B**) *E. coli* 8178 reduces the load of *S.* Tm in vivo. The *S.* Tm load is plotted and compared between *S.* Tm mono-infected and *S.* Tm + *E. coli* 8178 infected mice. (**C**) Influence of *E. coli* 8178 in *S.* Tm-elicited inflammation. The inflammation level is determined by measuring the host inflammation-associated marker lipocalin-2 (LCN2) from faecal samples of independently infected mice at different time

points. (D) Inflammation initiates *E. coli* 8178 –*S*. Tm competition. Competitive infection experiments using a combination of bacterial strains including *E. coli* 8178, *S*. Tm and a *S*. Tm mutant deprived of the main inflammation-triggering factors (Δ*spi*-1Δ*spi*-2). The strains used in each condition are depicted below. The *S*. Tm load collected 96 h after infection is presented. +: presence; -: absence. (**B–D**) The x-axis represents the time post-infection (in hours). Bars: median. Dotted line: limit of detection. c.f.u: colony forming units. Two-tailed Mann–Whitney U tests to compare 2 groups in each panel. $p \geq 0.05$ not significant (ns), $p < 0.05$ (*), $p < 0.01$ (**), $p < 0.005$ (***). The data underlying this figure can be found in S1 Data. p.i, post-infection; SPF, specific pathogen-free.

we rationally targeted metabolic genes that are known to play a significant role in *E. coli*'s competitiveness in the murine gut [12,14,16,29,30,37]. Given that *E. coli* 8178 competes against *S*. Tm in an inflammation-dependent manner, we hypothesized that mutations affecting *E. coli* 8178's competitiveness toward *S*. Tm would be disadvantageous during inflammation but neutral in a non-inflamed condition. We thus infected mice with a 1:1:1. . . mixture of *E. coli* 8178 mutants and WT, screening for mutations that were (1) neutral when competing against the *S*. Tm *spi*-1/*spi*-2 strain; and (2) attenuated in presence of the WT *S*. Tm (Fig 2A). The fitness of each individual mutant was assessed by quantitative PCR (qPCR) through the detection of a unique DNA barcode and calculated as a normalized competitive index (C.I) [38]. All *E. coli* 8178 mutants were found to stably colonize the murine gut as no growth defect was observed 24 h post-infection (S2A Fig). However, several mutants featured a pronounced and significant growth attenuation (up to 1,000-fold) 96 h post-infection (Fig 2B). In line with earlier work indicating the release of electron acceptors (different than fumarate) during *S*. Tm-elicited inflammation, an *E. coli* 8178 incapable of fumarate respiration (Δ*frdABCD*) exhibited a competitive disadvantage that was less pronounced in the inflamed than in the non-inflamed gut (Fig 2B) [23,39,40]. Conversely, the fitness of the *E. coli* 8178 *moaA*, *tonB*, and *cydAB* mutant was exclusively attenuated in the presence of the WT *S*. Tm strain, while remaining neutral in mice co-infected with the *S*. Tm *spi*-1/*spi*-2 mutant (Fig 2B). This led us to consider that either *E. coli* 8178 *moaA*, *tonB*, or *cydAB* was potentially playing a role in mediating *S*. Tm elimination in the inflamed gut. *cydAB* and *moaA* are respectively involved in the aerobic and molybdenum-dependent anaerobic bacterial respiration while *tonB* participates in the energy-dependent uptake of metabolites [16,37,41–43]. To directly assess the contribution of these bacterial processes in *S*. Tm gut competition, we infected mice with an equal mixture of *S*. Tm and the respective *E. coli* 8178 mutants. In contrast to *moaA* and *cydAB*, disruption of *tonB* in *E. coli* 8178 resulted in a higher survival of *S*. Tm in the mouse intestine (Fig 2C). However, this attenuated competitive effect was associated with a reduced *E. coli* 8178 *tonB* gut colonization, raising the alternative hypothesis that *tonB* deletion could indirectly affect *E. coli* 8178-mediated competition by impacting bacterial growth (S2B Fig). To establish a direct or indirect participation of *tonB*, we decided to further decipher its contribution in *S*. Tm elimination.

## *E. coli* 8178 salmochelin siderophore plays a pivotal function in *S*. Tm competition

TonB is an inner membrane protein facilitating the energy-dependent transport of several molecules [44,45]. Among these are vitamin B12 and iron-siderophore complexes, which were shown to transit through different *tonB*-dependent receptors (TBDRs) [43]. Notably, recent work in the literature demonstrated the contribution of the *E. coli* Nissle *tonB* gene in limiting *S*. Tm growth in vivo [16]. This effect was attributed to the incapacity of an *E. coli* Nissle *tonB* mutant to compete for iron, thereby allowing the bloom of *S*. Tm in the inflamed gut. Drawing from this example, we speculated that *E. coli* 8178 outgrows *S*. Tm through iron acquisition. To test this hypothesis and identify the specific *tonB*-related pathway responsible for *S*. Tm elimination, we tested a set of different TBDR mutants in *E. coli* 8178. First, we evaluated the

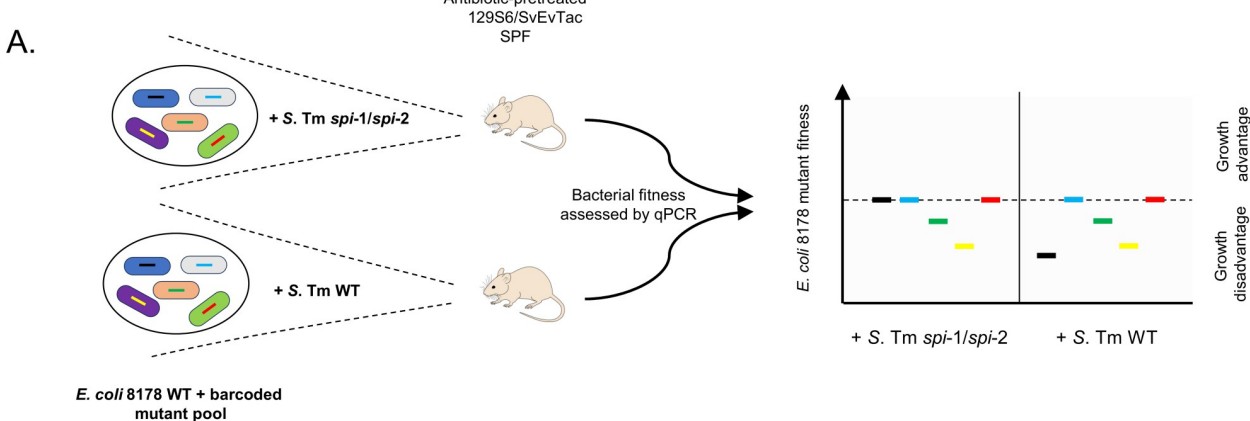

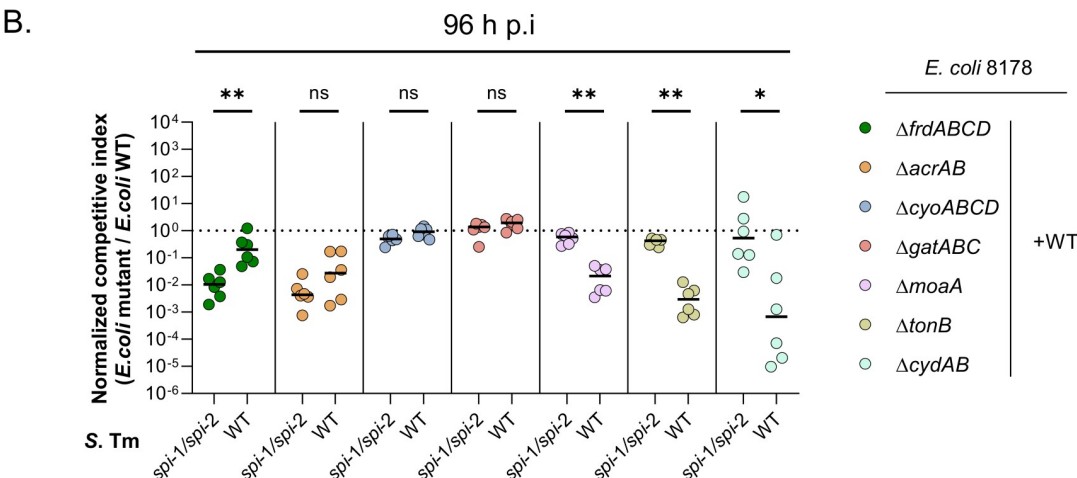

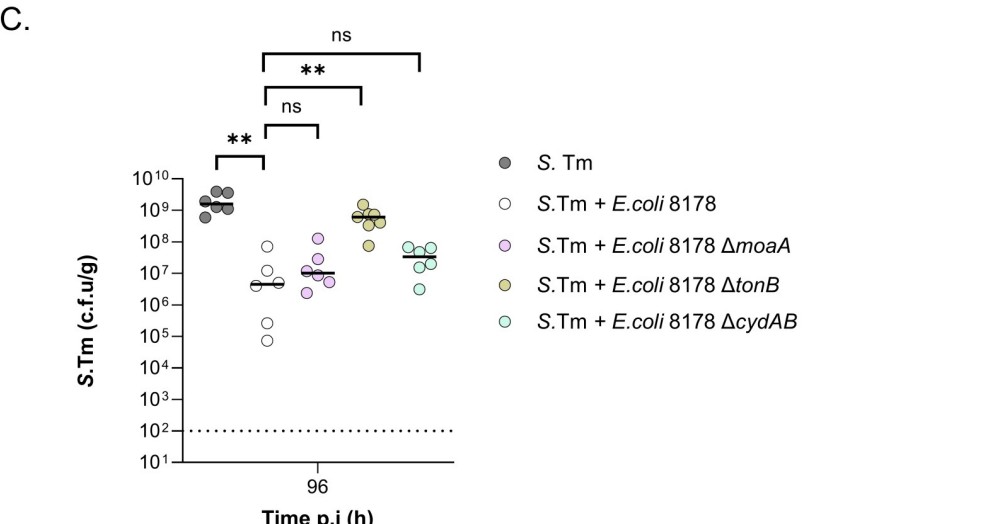

**Fig 2. *E. coli* 8178's TonB is required to effectively outcompete *S.* Tm in vivo.** (**A**) Screening strategy. Streptomycin-pretreated SPF 129S6/
SvEvTac mice were infected with the *E. coli* 8178 WT and a pool of rationally designed barcoded mutants. The fitness of the individual mutant was

compared to the WT strain by qPCR under conditions involving different *S.* Tm strains conditioning the competition outcome. Mock data are depicted in the graph. (**B**) Analysis of *E. coli* 8178's competitive factors in vivo. The normalized C.I of each individual *E. coli* 8178 mutant (listed) is determined 96 h p.i and plotted (y-axis) for each condition tested (x-axis). Dotted line: C.I expected for a fitness-neutral mutation. (**C**) Competitive infection experiments. The *S.* Tm load is plotted and compared between *S.* Tm mono-infected and S. Tm + *E. coli* 8178 infected mice. The *E. coli* 8178 mutant strains tested are listed. The x-axis represents the time post-infection (in hours). Dotted line: limit of detection. c.f.u: colony forming units. (**B, C**) Bars: median. Two-tailed Mann–Whitney U tests to compare 2 groups in each panel. $p \geq 0.05$ not significant (ns), $p < 0.05$ (*), $p < 0.01$ (**). The data underlying this figure can be found in S1 Data. C.I, competitive index; p.i, post-infection; qPCR, quantitative PCR; SPF, specific pathogen-free.

contribution of the vitamin B12 uptake pathway by creating an *E. coli* 8178 mutant deprived from the main *tonB*-dependent vitamin B12 uptake receptor, BtuB [46]. As expected, a *btuB* mutant was successfully outgrowing *S.* Tm in co-infection mice experiments, indicating that *E. coli* 8178 is not competing against *S.* Tm through vitamin B12 uptake (Figs 3A and S3A). Next, we systematically evaluated all TBDR genes involved in *E. coli* 8178 iron uptake, namely *cirA*, *chuA*, *fhuA*, *fhuE*, *fepA*, *fiu*, *iroN*, *fuyA* contributing to heme, catecholate-, yersiniabactin-, and hydroxamate- siderophores intake (S3B Fig) [47]. Due to the functional redundancy of the catecholate (*cirA*/*fepA*/*iroN*/*fiu*) and hydroxamate (*fhuE*/*fhuA*) siderophore receptors, we additionally deleted *fepB* and *fhuD* which respectively act downstream of the catecholate and hydroxamate uptake systems (S3B Fig) [48–50]. The resulting *E. coli* 8178 mutants were individually barcoded, pooled, and their fitness assessed by qPCR, with our primary focus on mutations that were (1) neutral when competing against the *S.* Tm *spi*-1/*spi*-2 strain; and (2) attenuated in presence of the WT *S.* Tm at 96 h post-infection (Figs 3B and S3C). The *E. coli* 8178 strains unable to uptake hemin (*chuA*), hydroxamate (*fhuA*, *fhuE*, *fhuD*), and yersiniabactin (*fuyA*) did not exert any fitness disadvantage in presence of inflammation, suggesting that none of these siderophores is responsible for *S.* Tm competition in that context. Disruption of the catecholate receptors *iroN*, *fiu*, and *fepA* attenuated the fitness of *E. coli* 8178 when grown in presence of the WT *S.* Tm, compared to the *S.* Tm *spi*-1/*spi*-2 mutant. This effect was further exacerbated in a *fepB* mutant which displayed a significant fitness defect exclusively in the inflamed gut, while having no growth disadvantage in a non-inflamed condition (Fig 3B). Altogether, these observations indicated that catecholate siderophores, which play a role in mediating *E. coli* 8178's survival under the iron-limited inflamed, may potentially contribute to *E. coli* 8178-mediated *S.* Tm competition. To directly test this hypothesis, we infected mice with a mixture of *S.* Tm and the *E. coli* 8178 *fepB* mutant. Compared to a WT strain, an *E. coli* 8178 *fepB* mutant has lost its ability to outgrow *S.* Tm in vivo (Figs 3C and S3D). We thus concluded that, similar to *E. coli* Nissle, *E. coli* 8178 relies on catecholate siderophore to outcompete *S.* Tm in the inflamed gut [16]. Catecholate molecules belong to a subfamily of siderophores that bind free iron through hydroxyl groups [51]. Enterobactin is a catecholate siderophore commonly found within the *Enterobacteriaceae* family. Beyond enterobactin, previous studies have shown that certain *S.* Tm and *E. coli* strains were able to produce salmochelin, a siderophore resulting from the glycosylation of enterobactin [52–54]. To further identify the catecholate siderophore responsible for *S.* Tm growth reduction, we initiated a genomic search for the enterobactin and salmochelin synthesis clusters and found that *E. coli* 8178 harbors both iron acquisition systems. Mutation of the *entA* gene abrogating the synthesis of enterobactin and thereby salmochelin attenuates *E. coli* 8178's ability to outperform *S.* Tm (Figs 3D and S3E). A similar trend was observed with an *iroB* mutant strain which, lacking its homolog *mchA*, is incapable of converting enterobactin into salmochelin (Figs 3D and S3E). Collectively, our data indicated that the catecholate siderophore salmochelin plays a critical role in providing *E. coli* 8178 with a competitive advantage over *S.* Tm.

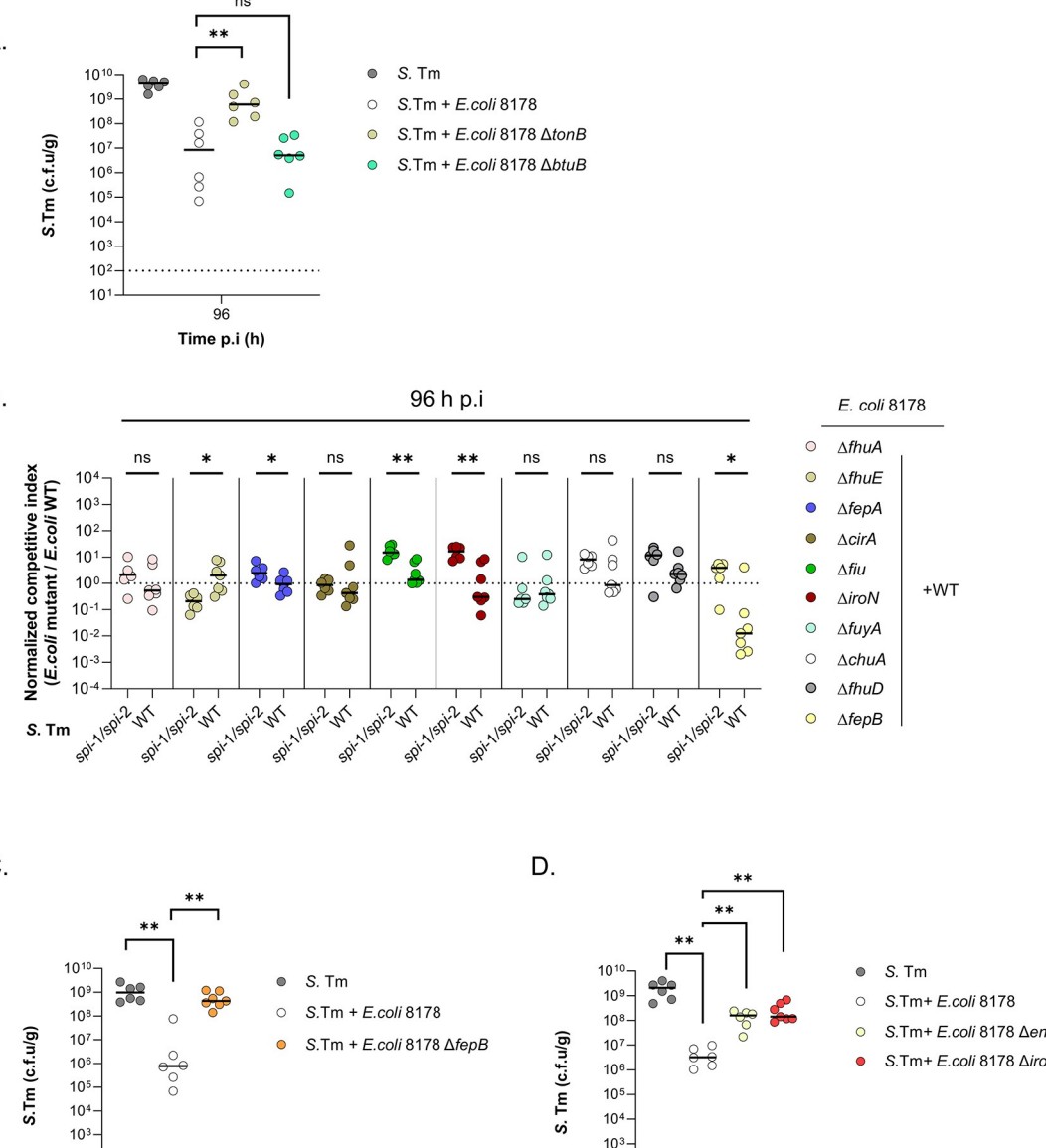

**Fig 3. *E. coli* 8178 relies on salmochelin siderophore to attenuate *S*. Tm growth in the gut.** (**A**) Competitive infection experiments. The *S*. Tm load is plotted and compared between *S*. Tm mono-infected and S. Tm + *E. coli* 8178 infected mice. The *E. coli* 8178 mutant strains tested are listed. (**B**) Analysis of *E. coli* 8178's TBDRs and siderophore uptake contribution in vivo. The normalized C.I of each individual *E. coli* 8178 mutant (listed) is determined 96 h p.i and plotted (y-axis) for each condition tested (x-axis). Dotted line: C.I expected for a fitness-neutral mutation. (**C, D**) *E. coli* 8178's catecholate siderophores are required to reduce *S*. Tm fitness in vivo. The *S*. Tm load is plotted and compared between *S*. Tm mono-infected and *S*. Tm + *E. coli* 8178 infected mice. The *E. coli* 8178 mutant strains tested are listed. The x-axis represents the time post-infection (in hours). Dotted line: limit of detection. (**A–D**) Bars: median. c.f.u: colony forming units. Two-tailed Mann–Whitney U tests to compare 2 groups in each panel. $p \geq 0.05$ not significant (ns), $p < 0.05$ (*), $p < 0.01$ (**). The data underlying this figure can be found in S1 Data. C.I, competitive index; p.i, post-infection; TBDR, *tonB*-dependent receptor.

## Microcin-bound salmochelins are produced by *E. coli* 8178 to outcompete *S.* Tm in the inflamed gut

Competition for iron is a common process through which microorganisms utilize optimized siderophore systems to outgrow their niche rivals [55]. This strategy is exemplified by certain *S.* Tm or *E. coli* strains that utilize salmochelin to acquire iron, thereby gaining an advantage over competing strains relying on enterobactin whose availability is limited by the host [54,56]. However, in the current case, a high similarity between their salmochelin gene clusters raised questions about how *E. coli* 8178 could more effectively benefit from salmochelin than *S.* Tm. We therefore suspected that *E. coli* 8178 does not rely on the iron acquisition property of salmochelin per se but instead utilizes an alternative killing mechanism dependent on salmochelin synthesis. Besides chelating iron, siderophores can act as cargo molecules for the transport of toxic proteins in a phenomenon known as the "trojan horse" strategy [57,58]. This is further illustrated by *E. coli* Nissle, which was shown to eliminate *S.* Tm through the secretion of siderophore-bound microcins in vivo [18]. A genomic search for antimicrobial gene clusters in *E. coli* 8178 revealed the presence of several toxins (colibactin, microcins) and antimicrobial contact-dependent machineries (type six-secretion system, contact-dependent growth inhibition). To investigate their contribution, we systematically deleted each interference system and individually tested the mutant's competitiveness against *S.* Tm (Fig 4A). With the exception of *mchB*, all mutants analyzed were effective in outcompeting *S.* Tm 96 h post-infection. The *mchB* gene encodes for the precursor of a low-molecular weight antibacterial toxin named microcin H47, which is fused to a salmochelin siderophore moiety and is exported as a microcin H47-bound salmochelin complex [59–61]. Notably, an *E. coli* 8178 deficient for (1) microcin H47 and salmochelin synthesis (*mchB/iroB*); or (2) microcin H47 synthesis and salmochelin intake (*mchB/iroN*), phenocopies a *mchB* mutant (Figs 4B and S4A). This confirmed that the growth attenuation of *S.* Tm by *E. coli* 8178 is solely dependent on microcin H47 synthesis rather than salmochelin acquisition. To demonstrate *E. coli* 8178's ability to produce microcin H47 in the inflamed gut, we created a plasmid-based reporter (*p-PmchI-luc*) expressing a luciferase gene under the control of the microcin H47 synthesis gene cluster promoter [62]. Inoculation of *S.* Tm infected mice with *E. coli* 8178 bearing the *p-.PmchI-luc* reporter resulted in a strong luciferase signal visible at 96 h post-infection, which was drastically reduced in animals experiencing no severe intestinal inflammation (Figs 4C and S4B). To further prove that inflammation acts as a cue to trigger microcin H47-dependent competition, and considering the difficulties associated with detecting microcin H47 from in vivo samples, we designed an *E. coli* 8178 *mchI/mchB* double mutant. This strain lacks the gene responsible for microcin immunity and synthesis, making it susceptible to microcin-dependent killing by the WT strain. Triple infection experiments with *S.* Tm along with the WT and *mchI/mchB E. coli* 8178 strains resulted in the attenuation of the microcin-sensitive mutant in a microcin H47- and inflammation-dependent manner (Fig 4D). Finally, *in trans* expression of the *mchI* immunity gene in *S.* Tm increased the pathogen's survival against *E. coli* 8178 (Figs 4E and S4C). Taken together, these data indicate that in contrast to *E. coli* Nissle that predominantly uses microcin M, *E. coli* 8178 produces microcin H47 in an inflammation-dependent manner to eliminate closely related competitors such as *S.* Tm in vivo [18].

Although these observations conclusively indicate microcin H47 as the primary contributor of *S.* Tm elimination, it remained unclear why an *E. coli* 8178 *tonB* and *fepB* mutant displayed a reduced capacity to attenuate *S.* Tm growth in vivo. To reconcile our findings, we investigated whether disruption of *fepB* and *tonB* gene might interfere with the microcin H47-dependent killing of *E. coli* 8178. For this aim, we established an in vitro killing assay and assessed

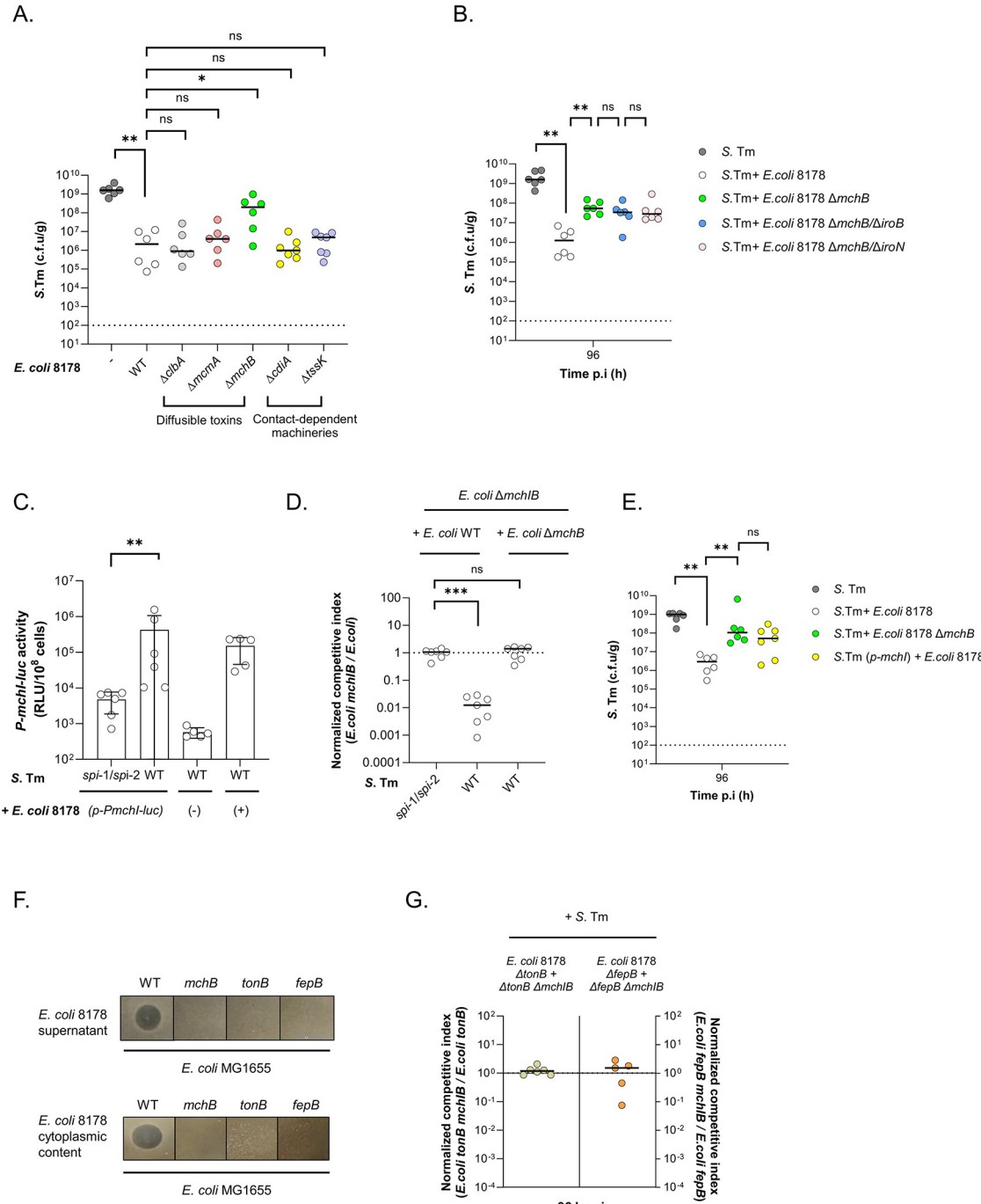

**Fig 4. *E. coli* 8178 produces microcin H47-bound salmochelin to effectively eliminate *S.* Tm in vivo.** (**A**) Individual contribution of *E. coli* 8178's interference systems against *S.* Tm. The faecal load of *S.* Tm collected 96 h p.i is plotted and compared between *S.* Tm mono-infected and S. Tm + *E. coli* 8178 infected mice. The *E. coli* 8178 mutant strains tested are listed in the x-axis and categorized based on the nature of the interference system disrupted. (**B**) Competitive infection experiments. The *S.* Tm load is plotted and compared between *S.* Tm mono-infected and S. Tm + *E. coli* 8178 infected mice. The *E. coli* 8178 mutant strains tested are listed. (**C**) *E. coli* 8178 produces microcin H47 in an inflammation-dependent manner. Microcin H47 expression is assessed using a plasmid-based transcriptional reporter (*p-PmchI-luc*). The luciferase signal, indicative of the microcin H47 gene expression level, is measured at 96 h p.i from faecal samples, both in the presence (+ *S.* Tm WT) and absence (+ *S.* Tm *spi*-1/*spi*-2) of inflammation. Negative and positive controls corresponding respectively to the luciferase signal from an empty (-) and a constitutively expressing luciferase plasmid (+) are also shown. RLU: relative light unit. The mean and standard deviation are represented. (**D**) Microcin H47-dependent killing is effective during inflammation. The fitness of the *E. coli* 8178 microcin-sensitive strain (Δ*mchIB*) is compared to the WT or microcin synthesis-defective strain (Δ*mchB*) in the presence (+ *S.* Tm WT) or absence (+

*spi*-1/*spi*-2) of inflammation. The normalized C.I of *E. coli* 8178 Δ*mchIB* is determined 96 h p.i and plotted (y-axis) for each condition depicted. Dotted line: C.I expected for a fitness-neutral mutation. (**E**) The microcin H47 immunity gene confers a higher *S*. Tm survival against *E. coli* 8178. The *S*. Tm load is plotted and compared between *S*. Tm mono-infected and *S*. Tm + *E. coli* 8178 infected mice. *S*. Tm (*p-mchI*): *S*. Tm constitutively expressing *mchI* immunity gene. (**F**) Disruption of the catecholate siderophore intake system impacts microcin H47 production in *E. coli* 8178. In vitro killing assay assessing the level of secreted (top) and intracellularly produced (bottom) microcin H47 in each *E. coli* 8178 strains listed. Microcin H47 activity killing is detected by the apparition of a lysis zone using *E. coli* MG1655 as a prey. (**G**) Deletion of *tonB* or *fepB* impairs microcin H47-dependent killing in vivo. The fitness of the microcin-sensitive *tonB* (Δ*tonB* Δ*mchIB*; left axis) and *fepB* (Δ*fepB* Δ*mchIB*; right axis) strains is compared respectively to the *E. coli* 8178 *tonB* and *fepB* mutant, in presence of inflammation (+ *S*. Tm). The normalized C.I of the microcin-susceptible *E. coli* 8178 mutants is determined at 96 h p.i. Dotted line: C.I expected for a fitness-neutral mutation. (**A, B, D, E, G**) Bars: median. c.f.u: colony forming units. Two-tailed Mann–Whitney U tests to compare 2 groups in each panel. $p \geq 0.05$ not significant (ns), $p < 0.05$ (\*), $p < 0.01$ (\*\*). (**A, B, E**) Dotted line: limit of detection. (**A–G**) The data underlying this figure can be found in S1 Data. C.I, competitive index; p.i, post-infection.

the level of microcin H47 synthesis in different *E. coli* 8178 mutant backgrounds under iron-deprived condition, a cue co-occurring with intestinal inflammation and known to trigger microcin production [18,61]. The cleared supernatant from stationary phase grown *E. coli* 8178 inhibited the growth of an *E. coli* K-12 prey strain in a microcin H47-dependent fashion (Fig 4F). However, upon incubating *E. coli* K-12 with the supernatant of an *E. coli* 8178 *tonB* or *fepB* mutant, no growth inhibition was observed (Fig 4F). This indicated an unexpected functional crosstalk between siderophore uptake and microcin H47 production and/or secretion. To further characterize this intertwined relationship, we assessed the microcin production capacity of different *E. coli* 8178 mutants by incubating their bacterial lysate with *E. coli* K-12 (Fig 4F). In comparison to the WT strain, the intracellular fraction of the *E. coli* 8178 *tonB* and *fepB* mutant did not inhibit *E. coli* K-12 growth, indicating a defect in the synthesis of active microcin H47 in these strains. To confirm the impaired microcin H47-dependent killing upon disruption of *tonB* and *fepB* from an in vivo perspective, we engineered an *E. coli* 8178 *tonB* and *fepB* mutant susceptible to microcin H47 (*E. coli* 8178 *tonB*/*mchB*/*mchI* and *fepB*/*mchB*/*mchI*) and analyzed their fitness under inflamed conditions. In line with our in vitro data, the *E. coli* 8178 *tonB* and *fepB* mutant failed to outcompete the microcin H47-sensitive *tonB* and *fepB* strains (Fig 4G). Production of active microcin H47 molecules involves a multistage process including the synthesis of the siderophore carrier, the microcin H47 precursor, and their maturation [59,60]. Using our plasmid reporter tool, we discovered that the expression of the microcin H47 synthesis cluster was comparable among *E. coli* 8178 WT, *tonB* and *fepB* strains under iron-limited condition (S4D Fig). This indicated that disruption of *tonB* and *fepB* may impact siderophore production and/or microcin maturation instead. Together, these data suggest that the attenuated *S*. Tm competition capacity of the *E. coli* 8178 *tonB* and *fepB* mutant is not attributed to iron acquisition but rather a defect in the formation of mature microcin H47 molecules.

## Discussion

Gut inflammation is associated with significant changes that are reflected at both the microbial community and environmental composition [20,21,63]. Upon infection, the host limits the availability of essential trace minerals, such as iron, to restrict bacterial proliferation. This phenomenon, commonly referred to as nutritional immunity, is ensured by the production of iron-scavenging and the lipocalin-2 enterobactin-sequestering proteins which respectively limit the availability of free- and bacterial siderophore bound-iron in the gut [64,65]. Bacteria can overcome host-mediated iron limitation through the synthesis of iron-rich ferrosome organelles, or the secretion of lipoproteins attaching to enterobactin, thereby preventing sequestration by the host lipocalin-2 [66,67]. Alternatively, certain members of the *Enterobacteriaceae* family can synthetize salmochelin, an enterobactin-derived siderophore that evades

lipocalin-2 attachment [52–54,56]. Apart from the ongoing battle for iron against the host, bacteria are also engaged in a competition with each other to gain a better access to the limited iron supply. In this iron tug-of-war, bacteria can specifically target niche competitors through the utilization of sophisticated contact-dependent secretion machineries (i.e., T6SS) or by re-purposing siderophore-scaffolds to deliver toxic molecules [18,68,69]. This holds true for *E. coli* 8178 which, despite encoding numerous antimicrobials systems, primarily relies on microcin H47 to outcompete *S.* Tm in infected 129S6/SvEvTac mice (Fig 4A). The preference for such diffusible toxins may be attributed to their long-range effect, which particularly bene-fit a population experiencing high loads, as *E. coli* 8178 encounters during gut inflammation (S1A Fig) [25,70]. However, it is important to note that *E. coli* and more generally members of the *Enterobacteriaceae* family are underrepresented in the healthy gut which consequently lim-its the average toxin concentration at the luminal side [71]. It remains yet to be determined whether diffusible molecules such as microcins can still provide benefits under these circumstances.

Microcins stand out as a narrow-spectrum toxin targeting bacteria from the *Enterobacteria-ceae* family [72]. Based on their size and structural characteristics, microcins can be subdivided into different classes. Microcin H47 is part of the class IIb family that carries a C-terminal post-translation modification involving a catecholate-siderophore moiety [61,72]. Such singu-larity makes microcin H47 a highly specific antimicrobial molecule conferring to the producer strain an exceptionally efficient strategy for eliminating closely related rivals relying on similar siderophores. As a result, microcin H47 was shown to exert an antibacterial activity against several members of the *Enterobacteriaceae* family in vitro, a finding now corroborated in vivo using the natural mouse commensal isolate *E. coli* 8178 [61,73]. Specifically, an *E. coli* 8178 mutant incapable of synthetizing the microcin H47 precursor, or the siderophore carrier, dis-played a decreased competitiveness against *S.* Tm (Figs 3D and 4A). Besides the mouse com-mensal *E. coli* 8178 which relies on microcin H47, the human probiotic *E. coli* Nissle was shown to predominantly utilize a different class IIb microcin family member, microcin M, to mediate the elimination of *S.* Tm ATCC 14028 from the mammalian gut [18]. Although the reasons why *E. coli* 8178 and Nissle utilize different microcins to target *S.* Tm serovars remain unclear, recent in vitro studies revealed a heterogenous target spectrum of class 2B microcins [74]. This suggests the existence of specific microcin/target pairings and hints at undiscovered resistance mechanisms that will be an interesting topic for future work. Apart from commensal bacteria, microcin synthesis loci are also found in uropathogenic *E. coli* (UPEC) strains and more globally enriched within the invasive *E. coli* B2 phylogroup, which highlights their gen-eral contribution in microbial antagonistic relationships [57,75,76].

In addition to hijacking siderophore synthesis for antimicrobial purposes, bacteria can also gain an advantage over their competitors through the uptake or iron. This has been illustrated in *E. coli* Nissle which was shown to outgrow *S.* Tm through the *tonB*-dependent uptake of siderophores [16]. Similarly, we found that *E. coli* 8178 requires *tonB* and more precisely the catecholate intake siderophore protein *fepB* to eliminate *S.* Tm. However, while studying the *E. coli* 8178 *tonB*- and *fepB*-contribution in *S.* Tm competition, we found that microcin H47 synthesis was abrogated in these backgrounds. This observation, coupled with the redundant effect of a microcin H47- and siderophore synthesis-deficient strain (*mchB*/*iroB*), or a micro-cin H47-synthesis and salmochelin uptake-deficient mutant (*mchB*/*iroN*), suggested that the lost competitiveness of the *E. coli* 8178 *tonB* or *fepB* mutant against *S.* Tm is not attributed to a defect in iron acquisition but rather to the incapacity of producing microcin H47 (Fig 4B, 4F and 4G). Mechanistically, disruption of *tonB* or *fepB* in *E. coli* 8178 did not directly impact the expression of the microcin H47 synthesis gene cluster (S4D Fig), suggesting a potential effect at the level of siderophore synthesis and/or the microcin H47 maturation step instead.

Alternatively, it is worth noting that *tonB* is involved in the uptake of various other substrates (i.e., carbohydrates, copper, nickel) while *fepB* function is not restricted to catecholate sidero-phore uptake but can be further expanded to virulence [77,78]. A defect in either of these genes may have a pleiotropic effect, which could account for our observation. Such crosstalk requires careful and mechanistic disentanglement to further expand the function of *tonB/fepB* in bacterial physiology but also avoid any potential misinterpretation.

## Methods

### Animals and ethic statement

Male and female 8 to 12 weeks old 129S6/SvEvTac (Jackson Laboratory) mice were randomly assigned to experimental groups and used in this study. Mice were held under SPF conditions in individually ventilated cages at the EPIC mouse facility of ETH Zurich. All animal experiments were reviewed and approved by Tierversuchskommission, Kantonales Veterinäramt Zürich under licence ZH158/2019, ZH108/2022, ZH109/2022 complying with the cantonal and Swiss legislation.

### Strains, media, and chemicals

All strains, plasmids, and oligonucleotides used in this study are listed in S1–S3 Tables. The *E. coli* 8178 strains originate from previous work published in [25] (accession number: NZ_JAEFCJ010000000). Bacterial strains were routinely grown in lysogeny broth (LB) supplemented or not with bactoagar (1%). Plasmids were maintained and mutants selected through antibiotic addition: streptomycin (100 μg/ml), ampicillin (100 μg/ml), kanamycin (50 μg/ml), and chloramphenicol (30 μg/ml). Gene deletions were achieved in *E. coli* 8178 using a modi-fied version of the lambda red recombinase-dependent one-step inactivation procedure [79]. Briefly, the antibiotic cassette (kanamycin for gene deletion, ampicillin for barcode insertion) was PCR-amplified using primer pairs carrying a 50-nucleotide extension homologous to the adjacent targeted region. Mutants were obtained through the electroporation of the PCR prod-uct into *E. coli* 8178 cells expressing the lambda red recombinase from the pSIM5 plasmid and incubated on selective media [80]. Gene deletion was confirmed by colony PCR. In a similar fashion, barcoded strains were created through the insertion of 40-bp DNA WISH (Wild-type Isogenic Standardized Hybrid) tags within a fitness-neutral region of the *E. coli* 8178's genome [38].

### Mouse infection experiments

The 8- to 12-weeks-old mice were orally pretreated with streptomycin (25 mg) 24 h before inoculation. *S.* Tm and *E. coli* cultures were grown on LB at 37°C for 4 h and washed twice with a phosphate-buffered saline solution (PBS: 137 mM NaCl, 2.7 mM KCl, 10 mM Na$_2$HPO$_4$, and 1.8 mM KH$_2$PO). Prior to colonization, *E. coli* strains were electroporated with the pRSF1010 plasmid from *Salmonella enterica* serovar Typhimurium SL1344 which confers streptomycin resistance [6,81]. Each mouse was orally given a single 50-μl dose containing approximately $5.10^7$ colony forming units (c.f.u) of an inoculum mixture composed of an equal ratio of the indicated strains. Faeces samples were collected 24 h and 96 h post-infection. Animals were euthanized by CO$_2$ asphyxiation at day 4 post-infection. Faecal samples were suspended in 1 ml PBS and homogenized using a TissueLyser (Qiagen). The bacterial load was determined by plating the suspension on MacConkey or LB agar supplemented with proper antibiotics.

## Fitness measurement of barcoded strains

Faecal *E. coli* cells were inoculated in 3 ml LB (37˚C, overnight) supplemented with ampicillin to select and enrich for living *E. coli* barcoded strains. The bacterial cells were pelleted and stored at −20˚C. DNA was extracted from thawed pellets using commercial kits (Qiagen Mini DNA) according to the manufacturer's instructions. The relative densities of the different barcodes were determined by real-time PCR quantification using tag-specific primers [38]. The obtained ratio was multiplied by the number of c.f.u recovered from selective plating to calculate the absolute loads of each tagged strain. The load of every single mutant strain was normalized to the inoculum and used to calculate the normalized C.I in order to compare their fitness to the WT (the C.I of a WT strain and respective mutant in a competitive infection was determined as a ratio between the c.f.u (mutant) and c.f.u (WT) divided by the ratio of both strains in the inoculum).

## Luciferase activity measurement

The expression of the microcin H47 synthesis gene cluster was assessed through the *p-PmchI-luc* reporter plasmid. Approximately $10^8$ *E. coli* 8178 cells carrying the *p-PmchI-luc* plasmid were isolated from freshly collected faecal samples and subsequently pelleted by centrifugation (5,000 rpm, 10′). Intracellular luciferase proteins were released by bacterial cells lysis through freeze–thaw cycles. The lysate was aliquoted in a 96-well plate and processed following the instructions provided with the Nano-Glo Luciferase Assay System kit (Promega). Luminescence levels were detected by the BioTek Synergy H1 (Agilent) plate reader. For in vitro growth cultures, the same procedure was achieved, with the exception that the luminescence signal collected was adjusted to the optical density (O.D).

## In vitro killing assay

The attacker *E. coli* 8178 and prey MG1655 strains were individually incubated at 37˚C, under agitation (160 rpm), in liquid LB supplemented with the iron chelator 2,2′-Bipyridyl (0.2 mM). Once the late stationary phase reached (O.D$_{600}$ ~ 3), the prey was diluted (final O.D: 0.003) into a freshly prepared LB agar solution (1% m/v agar) containing 0.2 mM of 2,2′- Bipyridyl. The presence of microcin H47 was monitored by incubating 5 μl of the 0.2 μm-filtered supernatant from the attacker strains onto the LB agar containing the prey mix, overnight at 37˚C.

## Lipocalin

The host Lipocalin-2 protein level was measured using the DuoSet ELISA Development kit (R&D Systems) from faeces samples homogenized in 1 ml of PBS.

## Histological procedures

Caecal tissues from infected mice were collected, embedded in O.C.T (Sakura Finetek, USA), snap frozen in liquid nitrogen, and stored at −80˚C. Cryosections (5 μm) were mounted on glass slides, air dried, and stained with hematoxylin and eosin (HE). Cecum pathology was evaluated blindly using the histopathological scoring scheme described previously [6,82].

## Molecular biology

Custom oligonucleotides were synthetized by *Microsynth* and are listed in S3 Table. PCRs were performed using the Phusion DNA polymerase (Thermo Scientific) and PCR products were purified using the Nucleospin Gel and PCR clean-up mini kit (Macherey-Nagel). The constitutively *mchI* expressing plasmid (*p-mchI*) and microcin expression reporter (*p-PmchI-*

*luc*) were constructed by restriction-free cloning (Gibson assembly, New England Biolabs) and verified by DNA sequencing (*Microsynth*). The QIAprep Spin Miniprep kit (Qiagen) was used to extract plasmid from bacterial pellets. Real-time PCR was performed using the FastStart Universal SYBR Green Master (Sigma-Aldrich) according to the manufacturer's instructions.

### Statistical analysis

The statistical analysis and data graphical representation were performed using GraphPad Prism 9.2.0 version for Windows (GraphPad Software, La Jolla California, USA, www.graphpad.com). When applicable, the unpaired Mann–Whitney U-test (comparison of ranks) was used to assess statistical significance when 2 groups were compared. *P* values $< 0.05$ were considered to indicate statistical significance.

### Supporting information

**S1 Data. Excel spreadsheet containing, in separate sheets for each figure, the underlying and individual numerical data used for Figs 1B–1D, 2B, 2C, 3A–3D, 4A–4G, S1A, S1B, S1C, S1D, S1E, S1F, S2A, S2B, S3A, S3C–S3E and S4A–S4D.**
(XLSX)

**S1 Fig. Impact of inflammation on *E. coli* 8178's ability to outgrow *S.* Tm.** (**A**) The load of *E. coli* 8178 in *S.* Tm + *E. coli* 8178 infected mice (from Fig 1B) is plotted on the y-axis over time (x-axis). (**B, C**) Pre-inoculating mice with *E. coli* 8178 results in similar *S.* Tm growth kinetics as in co-infected animals. The experiment scheme (B) and *S.* Tm loads in *E. coli* 8178 pre-inoculated mice (C) are shown. PBS: phosphate-buffered saline solution. (D) The *S.* Tm *spi-*1/*spi-2* mutant is unable to trigger inflammation in streptomycin-pretreated 129S6/SvEv-Tac animals. Top: Histopathological analysis of the caecal tissues at 96 h post-infection (p.i) from streptomycin pre-treated 129S6/SvEvTac animals that were independently infected with either the *S.* Tm WT (left) or *spi-*1/*spi-2* mutant (right). The hematoxylin and eosin (HE) stained caecal tissue sections were scored for edema in the sub-mucosa, polymorphonuclear leukocytes (PMNs) infiltration, reductions in the numbers of goblet cells and epithelial layer damages. Bottom: Representative images of HE-stained caecal tissue sections from mice infected with the *S.* Tm WT or *spi-*1/*spi-2* strain. Lu. = Lumen. S. E = Sub-mucosal edema. Ep. = Epithelium. Black arrow indicates epithelium gap. Scale bar = 100 μm. (**E, F**) Faecal lipoca-lin-2 (LCN2) (**E**) and *E. coli* 8178 (**F**) levels in mice infected with a combination of *E. coli* 8178, *S.* Tm WT, and *spi-*1/*spi-2* mutant (from Fig 1D). +: presence; -: absence. (**A, C–F**) The x-axis represents the time post-infection (in hours). p.i: post-infection. Dotted line: limit of detection. Bars: median. c.f.u: colony forming units. Two-tailed Mann–Whitney U tests to compare 2 groups in each panel. $p \geq 0.05$ not significant (ns), $p < 0.05$ (*), $p < 0.01$ (**). The data underlying this figure can be found in S1 Data.
(TIFF)

**S2 Fig. Screening for competitive factors of *E. coli* 8178 in vivo.** (**A**) Normalized competitive index of *E. coli* 8178 mutants analyzed in Fig 2B at 24 h post-infection (p.i). Dotted line: C.I expected for a fitness-neutral mutation. x-axis: *S.* Tm conditioning strain added. (**B**) Competitive infection experiments. Level of individual *E. coli* 8178 mutant collected in infected animals from Fig 2C. Dotted line: limit of detection. (**A, B**) Bars: median, two-tailed Mann–Whitney U tests to compare 2 groups in each panel. $p \geq 0.05$ not significant (ns), $p < 0.05$ (*), $p < 0.01$ (**). The data underlying this figure can be found in S1 Data.
(TIFF)

**S3 Fig. The uptake and synthesis of siderophore by *E. coli* 8178 contribute to *S.* Tm elimination.** (**A, D, E**) Level of individual *E. coli* 8178 mutant collected in infected animals from Fig 3A, 3C and 3D. p.i: post-infection. Dotted line: limit of detection. Bars: median. c.f.u: colony forming units. Two-tailed Mann–Whitney U tests to compare 2 groups in each panel. $p \geq 0.05$ not significant (ns), $p < 0.05$ (*), $p < 0.01$ (**). The data underlying this figure can be found in S1 Data. (**B**) Schematic representation of TBDRs and downstream actors facilitating siderophore uptake in *E. coli* 8178. (**C**) Normalized competitive index (C.I) of each individual *E. coli* 8178 mutant (listed) from Fig 3B, 24 h post-infection. Dotted line: C.I expected for a fitness-neutral mutation.
(TIFF)

**S4 Fig. *E. coli* 8178 relies on microcin H47 to outcompete *S.* Tm in the gut.** (A) Level of individual *E. coli* 8178 mutant collected in infected animals from Fig 4B and 4E. (**B**) The microcin H47 reporter plasmid does not impact *E. coli* 8178 growth in the mouse gut. Level of *E. coli* WT and *E. coli* (*p-PmchI-luc*) collected in *S.* Tm-infected animals at 24 h and 96 h post-infection. (**C**) Constitutive expression of the microcin H47 immunity (*mchI*) does not impact *S.* Tm fitness in the gut. The competitive index of the *mchI*-expressing *S.* Tm (*S.* Tm *p-mchI*) mutant relative to the WT strain is assessed at 24 h and 96 h post-infection, in absence of *E. coli* 8178. Dotted line: C.I expected for a fitness-neutral mutation. (**D**) Deletion of *tonB* of *fepB* does not impair expression of the microcin H47 synthesis gene cluster. Microcin H47 expression is evaluated using a plasmid-based transcriptional reporter (*p-PmchI-luc*) in the *E. coli* 8178 WT, *ΔtonB* and *ΔfepB* strains, after 8 h incubation in an iron-deprived medium. Negative and positive controls corresponding respectively to the luciferase signal from an empty (-) and a constitutively expressing luciferase plasmid (+) are also shown. RLU: relative light unit. The mean and standard deviation of $n = 5$ biological replicates are represented. (**A–C**) p.i: post-infection. Bars: median. (**A–D**) Two-tailed Mann–Whitney U tests to compare 2 groups in each panel. $p \geq 0.05$ not significant (ns), $p < 0.05$ (*), $p < 0.01$ (**). The data underlying this figure can be found in S1 Data.
(TIFF)

**S1 Table. Strains, primers, and plasmids used in this study.**
(XLSX)

**S2 Table. Strains, primers, and plasmids used in this study.**
(XLSX)

**S3 Table. Strains, primers, and plasmids used in this study.**
(XLSX)

**S1 Raw Images. Uncropped pictures used in Fig 4F.**
(PDF)

## Acknowledgments

We would like to acknowledge the staff at the ETH animal facilities (EPIC and RCHCI; especially Manuela Graf, Katharina Holzinger, Dennis Mollenhauer, Sven Nowok, and Dominik Bacovcin) and thank members of the Hardt lab for helpful comments. We thank Andrew Abi Younes for his feedback on the manuscript, as well as Manja Barthel, Luca Maurer, and Ursina Enz for their assistance with tissue sectioning and staining. Valuable help from the ETH students Jasmin Baumgartner, Diana Solenthaler, and Lorenz Kaiser was appreciated.

## Author Contributions

**Conceptualization:** Yassine Cherrak, Wolf-Dietrich Hardt.

**Data curation:** Yassine Cherrak.

**Formal analysis:** Yassine Cherrak, Miguel Angel Salazar, Markus Kreuzer.

**Funding acquisition:** Wolf-Dietrich Hardt.

**Investigation:** Yassine Cherrak, Miguel Angel Salazar, Koray Yilmaz, Markus Kreuzer, Wolf-Dietrich Hardt.

**Methodology:** Yassine Cherrak.

**Project administration:** Wolf-Dietrich Hardt.

**Resources:** Wolf-Dietrich Hardt.

**Supervision:** Yassine Cherrak, Wolf-Dietrich Hardt.

**Validation:** Yassine Cherrak.

**Writing – original draft:** Yassine Cherrak.

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
