## [Editor Report · Decision Letter 0]

19 Dec 2023

Dear Dr Hardt, 

Thank you for submitting your manuscript entitled "A commensal E. coli strain limits Salmonella gut invasion by producing toxin-bound siderophores in a tonB-dependent manner." for consideration as a Research Article by PLOS Biology. 

Your manuscript has now been evaluated by the PLOS Biology editorial staff [as well as by an academic editor with relevant expertise and I am writing to let you know that we would like to send your submission out for external peer review as a Short Report. 

However, before we can send your manuscript to reviewers, we need you to complete your submission by providing the metadata that is required for full assessment. To this end, please login to Editorial Manager where you will find the paper in the 'Submissions Needing Revisions' folder on your homepage. Please click 'Revise Submission' from the Action Links and complete all additional questions in the submission questionnaire. Please, when adding the rest of the metadata choose "Short Report".

Once your full submission is complete, your paper will undergo a series of checks in preparation for peer review. After your manuscript has passed the checks it will be sent out for review. To provide the metadata for your submission, please Login to Editorial Manager (https://www.editorialmanager.com/pbiology) within two working days, i.e. by Dec 21 2023 11:59PM.

Please notice that because of the incoming holidays, the peer-review process might take longer.

Kind regards,

Melissa

Melissa Vazquez Hernandez, Ph.D.

Associate Editor

PLOS Biology

---

## [Decision Letter · Decision Letter 1]

26 Jan 2024

Dear Dr Hardt,

Thank you for your patience while your manuscript "A commensal E. coli strain limits Salmonella gut invasion by producing toxin-bound siderophores in a tonB-dependent manner" was peer-reviewed at PLOS Biology. It has now been evaluated by the PLOS Biology editors, an Academic Editor with relevant expertise, and by four independent reviewers, one of whom signed their report and is Vanni Bucci.

In light of the reviews, which you will find at the end of this email, we would like to invite you to revise the work to thoroughly address the reviewers' reports. As you will see below, all reviewers are quite positive and interested in the work but agree that some additional experiments would aid in the conclusiveness of the work. Overall, there clearly is sufficient interest for us to consider the study further, but some revision will be necessary for publication in PLOS Biology. Reviewer 1 raises concerns regarding the validation of the mutant fitness, and reviewers 2 and 3 would like to see experimentally if microcin production is affected. Additionally, reviewer 3 would like to see experiments demonstrating that inflammation regulates the production of microcin-H47, while reviewer 4 would like to see some clarification in conclusions that may not be fully supported by the data presented. Finally, all reviewers consider that the study should better contextualize the findings with respect to previous literature. We agree with all reviewer concerns and would require experimental revisions to address them, as we consider that this would strengthen the work.

Given the extent of revision needed, we cannot make a decision about publication until we have seen the revised manuscript and your response to the reviewers' comments. Your revised manuscript may be sent for further evaluation by all or a subset of the reviewers.

We expect to receive your revised manuscript within 3 months, however please let us know if you would require some more time for revision, which would not be a problem. Please email us (plosbiology@plos.org) if you have any questions or concerns, or would like to request an extension.

**IMPORTANT - SUBMITTING YOUR REVISION**

*Re-submission Checklist*

*Published Peer Review*

*PLOS Data Policy*

*Blot and Gel Data Policy*

Sincerely,

Melissa

Melissa Vazquez Hernandez, Ph.D.

Associate Editor

PLOS Biology

REVIEWER'S COMMENTS:

— — — — — 

Reviewer #1: 

A great study. Beautiful figures. I just have issues with the narrative and lack of contextualization with existing work. I hope you agree some changes along these lines might allow this work to be more useful to the field.

General comments:

I find the experimental design, hypothesis generating and testing, and conveying of results through figures to be really superb. It is a genuinely well thought-out and organized study. That said, I find that many of the results presented have previously been demonstrated in the literature, or are obvious implications of existing literature, without proper acknowledgement of that fact by the authors.

The manuscript largely focuses on the strain E. coli 8178, that appears to inhibit Salmonella via the same mechanisms already reported in the much more widely studied E. coli Nissle 1917 strain. There are many nuanced differences between the present study and the existing literature that a specialist in this area might be very interested to read. However, these differences are not discussed.

I trust the authors know the existing literature well, and I think a re-write of some of the narrative needs to give proper credit to past work, and better situate the current findings in light of published works. The final section of results (nor the discussion) not contextualizing the outcomes in the framework of Sassone-Corsi et al 2016 Nature, is a serious shortcoming. The authors do cite this paper and discuss it very briefly in the discussion, but the fact that these results are not really contextualized in the light of that study is very confusing. I think it would be much more beneficial to the scientific community if this work were to make an effort to demonstrate how it builds on previous research in this area, rather than largely ignore it. I know there are plenty of differences in the experimental design between Sassone-Corsi and the study at hand. Method of inducing inflammation a key difference between them. But this kind of contextualization which would be useful for the field is completely absent. 

The finding that tonB and fepB mutations result in a loss of detectable microcin production in the assay performed here is indeed an interesting and perhaps unexpected result. I don't know if this phenomenon has previously been reported in the literature. However, this result is left as mostly an aside and not explored in much detail. 

Major Comments

Line 147 - The validation of the method described seems to rely exclusively on the Benjamin et al paper which is in revision. Either the methods used for this aspect of the work should be described and validated in greater detail, or this manuscript should be published only once the Benjamin paper passes peer review. Additionally, are these targeted mutations or random? If targeted, how were these sites for mutagenesis selected? 

1) Results Lines 168 - 215: The role of TonB in the uptake of both siderophores and siderophore-iron complexes is extremely well-established. This section seems written without that fact in mind. While Deriu 2013 is cited at the start, the authors seem to suggest that the results of that work only demonstrate that uptake of siderophores led EcN's ability to effectively compete with S. Tm. They then state tonB has a pleiotropic effect regarding siderophore iron complexes (again, the connection is extremely well-established). Deriu 2013 clearly state that their results suggest that it is the competition for iron (via siderophore-iron complexes) which leads to EcN-mediated competition with S. Tm. I find that this results section is written as though that is not a clear implication of Deriu 2013's data. All of the data provided are completely valid and clearly directly in line with the results of Deriu 2013's data. I don't understand why it is not written as such. 

Additionally, the focus on FepB is confusing. The essential role of FepB in catecholate siderophore uptake (and catecholate siderophore-iron complexes) is clearly established. This all again confirms clear expectations for existing literature. To suggest Ec 8178 specifically relies on FepB to compete with S. Tm just confuses the larger point, that your data is further confirmation that 8178 competes with S. Tm for iron via catecholate siderophores, which is well-established.

The authors have great experiments and very clear data and really nice figures. I just think the narrative of this results section is currently written in a way that mostly ignores what is known and expected based on existing literature, which has the potential to confuse readers. 

2) Line 239 - Many strains which harbour the Class IIB microcin production cluster also encode for a homolog of IroB, which is frequently identified as MchA. It glycosylated enterobactin into salmochelin, just as IroB does. Does 8178 have MchA? Without knowing this information, I cannot assess whether or not the claim of Line 239 is reasonable or not. If it is true that the iroB deletion mutant cannot make salmochelin, the claim should still be amended to state that 'attenuation of S. Tm by E. coli 8178 is solely dependent on microcin H47 synthesis rather than salmochelin acquisition.' You could also confirm what siderophores are present experimentally, using biochemical methods rather than relying on genetics. 

3) Results Line 216 - 266: The lack of reference to Sassone-Corsi et al Nature 2016 in this section is confusing. Again, these results seem to hold very well in light of that work. Granted, it seems here the effect is largely due to Microcin H47, rather than microcin M, in regards to the inhibitory effect against a strain of S. Tm. Though, in vitro tests have demonstrated the inhibitory potential of microcin H47 against strains of S. Tm (Palmer et al, 2020). 

4) Comparing figure 3A and 4D, if the primary cause of S. Tm inhibition is indeed due to Microcin H47, then I would expect a 8178 tonB mutant and a mchB mutant to result in similar levels of S. Tm (based on your experiments showing that TonB mutants are not producing microcins). However these seem to have roughly a 10-fold difference, where S. Tm in competition with an 8178 tonB mutant are doing ~10x better than compared to competition with a mchB mutant. How do you account for these differences? It seems to imply there is likely competition over both consuming iron and via microcins, which is not reflected in the interpretation of results. 

5) Figure 4 has me somewhat confused. 4A, 4B, and 4D all have the same X and Y axis and the same experimental set-up, but the data-points are clearly different. Taking the 8178 mchB mutants for example, why aren't all the data points pooled, or used across the different comparisons that you wish to show? Perhaps this is a nuance of high-throughput animal work that I'm unfamiliar with. It seems like an unfortunate waste of mice to be essentially repeating the same experiment over and over (e.g. S.Tm alone, S.Tm vs 8178 WT; S.Tm vs 8178 mchB_KO) for each plot subpanel.

minor comments

The abstract seems to imply that the OM receptors of Microcin H47 being TonB-dependent catecholate receptors was unappreciated, though this is well established in the literature. Clear experiments demonstrate this in Patzer et al 2003 - https://doi.org/10.1099/mic.0.26396-0. 

Line 72 - Is the co-blooming necessarily due to strong metabolic overlap between these species? Is this demonstrated in the referenced literature for this sentence? Could it also simply be due to things like increase in oxygen?

Figure 1 - It is hard to see the black mean lines associated with the black data points. Perhaps consider using data points which are somewhat lighter (more gray) in order for the black line to be observable. 

Line 111 - I don't understand the logic here. At 24 p.i., inflammation levels are reduced when co-infected, relative to S. Tm alone. Doesn't this imply some competition at 24hr p.i.? While it may not be the inflammation-induced competition mechanisms, it seems like it would indicate there is competition for nutrients, space, etc., reducing S. Tm's ability to cause inflammation.

Figure 2A - The fitness plots are unnecessary here. Either denote clearly in the caption that this is mock data for the purpose of demonstrate, or exclude this aspect of the experimental schematic.

Line 171 -To say that tonB has a pleiotropic function where, not only is it involved in siderophore uptake, but also the uptake of siderophore-iron complexes seems a bit misleading. This function of TonB is well-established.

Line 248 - From your own experiment, as well as experiments from Sassone-Corsi at al 2016, E. coli can clearly use these microcins to kill other E. coli as well. So it seems inappropriate to state that 8178 produces H47 to kill S. Tm. In the experimental design where S. Tm is present, yes that is agreed upon, but it seems to target any susceptible species present under inflammation conditions. While Salmonella is a common cause of inflammation, it is by no means the only. 

Line 287 -Indicate the type of filter used for the supernatant assays. Also indicate the volume of filtered supernatant deposited onto the agar.

— — — — — 

Reviewer #2 (Vanni Bucci): 

In this paper, Cherrak et al. present a comprehensive exploration of the mechanisms underlying the colonization resistance exerted by commensal E. coli strain 8178 against Salmonella Typhimurium (STm). Through several in vivo experiments, the authors reveal that E. coli 8178, in the inflamed gut, outcompetes STm by producing microcin H47. Notably, this study in a notable advancement as, until now, microcin H47-mediated inhibition of STm had only been demonstrated through in vitro assays (Palmer et al., 2018; Palmer et al

---

## [Editor Report · Decision Letter 2]

3 Apr 2024

Dear Dr Hardt,

Thank you for your patience while we considered your revised manuscript "A commensal E. coli strain limits Salmonella gut invasion during inflammation by producing toxin-bound siderophores in a tonB-dependent manner." for publication as a Short Reports at PLOS Biology. This revised version of your manuscript has been evaluated by the PLOS Biology editors, and the Academic Editor.

Based on our Academic Editor's assessment of your revision, we are likely to accept this manuscript for publication, provided you satisfactorily address the following remaining points. Please also make sure to address the following data and other policy-related requests.

a) We request the title to be changed to "Commensal E. coli limits Salmonella gut invasion during inflammation by producing toxin-bound siderophores in a tonB-dependent manner" — just removing the initial "A" and "strain"

b) Please cite the location of the data clearly in all relevant main and supplementary Figure legends (Fig. 1BCD, 2BC, 3ABCD, 4ABCDEG, S1ACDEF, S2AB, S3ACDE, S4ABCD), e.g. “The data underlying this Figure can be found in S1 Data”. A more detailed explanaition can be found after my signature.

c) The Ethics statement needs to be the first subheading in the Methods sections the Material & Methods section. You currently have it after “Strains, media and chemicals”; please move it before this section.

d) Instead of "Experimental model" and "Method detail subheading", please have all of this information under a "Methods" or "Material & Methods" subheading.

e) Please also see our code policy at the end of the e-mail

We expect to receive your revised manuscript within two weeks. 

*Published Peer Review History*

*Press*

Sincerely,

Melissa

Melissa Vazquez Hernandez, Ph.D.

Associate Editor

PLOS Biology

DATA POLICY:

[Fig. 1BCD, 2BC, 3ABCD, 4ABCDEG, S1ACDEF, S2AB, S3ACDE, S4ABCD.]

CODE POLICY

Per journal policy, if you have generated any custom code during the curse of this investigation, please make it available without restrictions upon publication. Please ensure that the code is sufficiently well documented and reusable, and that your Data Statement in the Editorial Manager submission system accurately describes where your code can be found. [IF APPLICABLE: As the code that you have generated to XXX is important to support the conclusions of your manuscript, its deposition is required for acceptance.]

---

## [Editor Report · Decision Letter 3]

5 Apr 2024

Dear Dr. Hardt,

Thank you for the submission of your revised Short Reports "Commensal E. coli limits Salmonella gut invasion during inflammation by producing toxin-bound siderophores in a tonB-dependent manner." for publication in PLOS Biology. On behalf of my colleagues and the Academic Editor, Sebastian E. Winter, I am pleased to say that we can in principle accept your manuscript for publication, provided you address the issues highlighted in the numbered points below. In addition, our colleagues in the journal operations team will send you an email within 2-3 business days asking you to address any other remaining formatting and reporting issues; no action is required from you until then so that you can address everything at one time. Please note that we will not be able to formally accept your manuscript and schedule it for publication until you have completed the requested changes.

1) The section called “Resource availability” with the sub-sections “Lead contact”, “Material availability” and “Data availability” needs to be deleted from the final version of the manuscript. The relevant information present in these sections should be added to the “Data Availability” section of your submission in Editorial Manager, as this will be published with the paper. It is self-understood that the corresponding author will be the contact for requests, so we do not need this explicitly stated, and the origin of the mice is already indicated in the Methods section.

2) During your submission, you have ticked “Tick here if the URLs/accession numbers/DOIs will be available only after acceptance of the manuscript for publication so that we can ensure their inclusion before publication." However, there does not seem to be anything deposited in databases. If this is the case, please untick this box during resubmission.

PRESS

Sincerely,

Melissa

Melissa Vazquez Hernandez, Ph.D.

Associate Editor

PLOS Biology
